# Targeted calcium influx boosts cytotoxic T lymphocyte function in the tumour microenvironment

Kyun-Do Kim[1], Seyeon Bae[1], Tara Capece[1], Hristina Nedelkovska[1], Rafael G. de Rubio[2], Alan V. Smrcka[2], Chang-Duk Jun[3], Woojin Jung[4], Byeonghak Park[4], Tae-il Kim[4] & Minsoo Kim[1]

Adoptive cell transfer utilizing tumour-targeting cytotoxic T lymphocytes (CTLs) is one of the most effective immunotherapies against haematological malignancies, but significant clinical success has not yet been achieved in solid tumours due in part to the strong immunosuppressive tumour microenvironment. Here, we show that suppression of CTL killing by $CD4^+CD25^+Foxp3^+$ regulatory T cell (Treg) is in part mediated by TGFβ-induced inhibition of inositol trisphosphate ($IP_3$) production, leading to a decrease in T cell receptor (TCR)-dependent intracellular $Ca^{2+}$ response. Highly selective optical control of $Ca^{2+}$ signalling in adoptively transferred CTLs enhances T cell activation and IFN-γ production in vitro, leading to a significant reduction in tumour growth in mice. Altogether, our findings indicate that the targeted optogenetic stimulation of intracellular $Ca^{2+}$ signal allows for the remote control of cytotoxic effector functions of adoptively transferred T cells with outstanding spatial resolution by boosting T cell immune responses at the tumour sites.

[1] Department of Microbiology and Immunology, David H. Smith Center for Vaccine Biology and Immunology, University of Rochester, Rochester, New York 14642, USA. [2] Department of Pharmacology & Physiology, University of Rochester, Rochester, New York 14642, USA. [3] School of Life Sciences, Immune Synapse and Cell Therapy Research Center, Gwangju Institute of Science and Technology, Gwangju 500-712, Republic of Korea. [4] School of Chemical Engineering, Sungkyunkwan University, Suwon, Gyeonggi-do 440-746, Republic of Korea. Correspondence and requests for materials should be addressed to M.K. (email: minsoo_kim@urmc.rochester.edu).

T cell-based immunotherapy has emerged as a powerful treatment option for several types of cancer[1]. However, this success has not yet been transferred to solid tumours, due in part to the strong immunosuppressive tumour microenvironment[2–6]. Systemic or intratumoural delivery of an immune-boosting molecule to overcome local suppression has been proposed, but the full potential of this approach is limited by non-specific stimulation of tumour growth, metastasis and angiogenesis[7–9].

During normal immune responses, Tregs suppress T cell effector functions by generating immunosuppressive adenosine, cyclic AMP (cAMP), or anti-inflammatory cytokines (IL-10, TGF-β, IL-35), and by consuming IL-2 (refs 10,11). Furthermore, Tregs can cause effector T cell death via granzyme and perforin, and suppress activation of T cells by downregulating costimulatory molecules on antigen presenting cells (APCs) via CTLA-4 (refs 10,11). In addition to this indirect regulation, Tregs have been shown to directly impair CD8[+] T cell effector functions by compromising the release of lytic granules on recognition of antigens on target cells[12,13]. Although, active regulation by Tregs plays a critical role in modulating host immunity, growing evidence suggests that FoxP3[+] Tregs negatively affect overall survival in the majority of solid tumours[14]. In particular, decreased ratios of CD8[+] T cells to Foxp3[+] Treg cells among tumour-infiltrating lymphocytes have been directly correlated with poor prognosis in ovarian, breast and gastric cancers[15–17]. Despite its key regulation of anti-tumour immune responses, the molecular mechanisms underlying Treg-mediated immune suppression in the tumour microenvironment are unclear. In this study, we used an optogenetic approach to control cytotoxic functions of CTLs with light stimulation. Our study revealed that highly selective optical control of Ca$^{2+}$ signalling in adoptively transferred CTLs was sufficient to overcome Treg-mediated immunosuppression at the tumour site, leading to a significant reduction in tumour growth in the mouse melanoma model.

## Results

**Suppression of CTL-mediated cytotoxicity by Tregs.** To determine the mechanisms of Treg-mediated immune suppression at the tumour site, we first measured the kinetics of Treg responses in a mouse melanoma model. C57BL/6 mice injected intradermally in the ear with B16F10 tumour cells developed solid tumours with steady increases in both absolute numbers and ratios of Foxp3[+]CD4[+] Treg cell population (Fig. 1a; Supplementary Fig. 1). Flow cytometry analysis revealed that the majority of Tregs in the tumour showed activated and terminal effector Treg phenotypes (CD25[high]ICOS[high]CTLA-4[high18]; Fig. 1b). The increased effector Treg cell counts in the tumour implied that these cells might play a key role in the loss of concomitant tumour immunity.

When exposed to target tumour cells, CD8[+] cytotoxic T cells (CD8[+] Tc) directly release cytotoxins, including perforin, granzymes and granulysin. Granzymes enter the cytoplasm of the tumour cell through a pore formed by perforin, and the serine protease function activates caspase cascades leading to tumour cell apoptosis. Therefore, the release of granule contents accounts for the most direct and final step in CD8[+] Tc effector function. Although multiple mechanisms underlying Treg-mediated immune suppression have been proposed[19], little is known about the role of Treg in regulation of the direct tumour killing process of CD8[+] Tc. To address this, we first seup an *in vitro* three cell killing assay, where OVA-loaded murine lymphoma EL-4 cells were co-cultured with CD8[+] Tc prepared from OT-I T cell receptor (TCR) transgenic mice in the presence of resting naive (rTreg) or activated effector (aTreg) Tregs. CD8[+] Tc alone displayed strong cytotoxicity (Annexin-V[+] or Propidium iodide[+]) against peptide-pulsed EL-4 (Fig. 1c). Preincubation of CD8[+] Tc with aTreg for 16 h completely abolished the tumouricidal functions of CD8[+] Tc, while incubation with rTreg had a lesser effect on the levels of cytotoxicity (Fig. 1c).

Importantly, expression of key effector molecules that directly induce CD8[+] Tc-mediated tumour killing, such as perforin and granzyme B, was not changed by co-incubation of CD8[+] Tc with aTreg (Fig. 1d). Instead, the impaired cytotoxicity was mainly associated with a decrease in granule exocytosis as measured by surface expression of CD107a (Fig. 1e). First, we suspected that the observed suppression of granule exocytosis and cytotoxic functions of CD8[+] Tc could be attributed to the Treg-mediated inhibition of the TCR itself or TCR-proximal signals (Fig. 1f). However, rapid tyrosine phosphorylation of CD3ζ in OT-I CD8[+] Tc on incubation with OVA-loaded EL-4 cells was not suppressed by co-incubation with aTreg (Fig. 1g). In addition, we detected similar levels of ZAP-70 phosphorylation in CD8[+] Tc both in the absence and presence of aTreg (Fig. 1g). The granule-mediated target cell killing of CD8[+] Tc is strictly calcium-dependent and requires store-operated Ca$^{2+}$ entry (SOCE)[20–22]. Orai1 and stromal interaction molecule 1 (STIM1) were identified as the molecular constituents of the calcium release-activated calcium (CRAC) channel in T cells (Fig. 1f)[23,24]. Therefore, we next turned our attention to T cell store-operated Ca$^{2+}$ entry activity and assessed whether Tregs suppress CD8[+] Tc lytic granule exocytosis by directly down-regulating Orai1 and/or STIM1 expression. Again, co-incubation of CD8[+] Tc with aTreg did not affect Orai1 and STIM1 expression levels (Fig. 1g). These results suggest that Tregs have a minimal impact on TCR activation and CRAC expression.

TCR activation induces hydrolysis of phosphatidylinositol-(4,5)-bisphosphate into inositol-(1,4,5)-trisphosphate (IP$_3$) by PLCγ, which induces the release of Ca$^{2+}$ from ER stores by activating IP$_3$-receptor (Fig. 1f). However, Tregs did not significantly change IP$_3$-receptor expression in CD8[+] Tc (Fig. 1h, left). Surprisingly, Tregs caused a significant decrease in TCR-induced IP production in CD8[+] Tc (Fig. 1h, right), which led to a dramatic reduction of both TCR (first peak)- and ionomycin (second peak)-induced intracellular Ca$^{2+}$ responses in CD8[+] Tc (Fig. 1i) and NFAT1 dephosphorylation (an effector molecule downstream of Ca$^{2+}$ signals in T cells) (Fig. 1j). Earlier studies reported that Treg cells directly suppress tumour-specific CD8[+] T cell cytotoxicity through TGFβ signals[25,26]. Importantly, it was shown that TGFβ suppresses Ca$^{2+}$ influx in activated T cells in part through the inhibition of interleukin-2 tyrosine kinase (ITK)-mediated PLCγ activation[27,28]. Similarly, aTreg-mediated suppression of CD8[+] Tc anti-tumour cytotoxicity was significantly decreased by the TGFβ superfamily type I activin receptor-like kinase receptor inhibitor SB431542 (Fig. 1k), suggesting that the Treg-mediated suppression of tumour killing through intracellular Ca$^{2+}$ signals is, at least in part, TGFβ-dependent.

**Ca$^{2+}$ signal and CD8[+] T cell cytotoxic functions.** The finding that Tregs directly inhibit the TCR-dependent granule exocytosis and tumouricidal functions of CD8[+] Tc by suppressing IP$_3$ production, and Ca$^{2+}$ influx suggests that strong intracellular Ca$^{2+}$ signals in CD8[+] Tc can enhance release of cytotoxic granules and thus boost CTL functions at tumour sites. To study the effects of increased intracellular Ca$^{2+}$ on T cell effector functions, we used the well-characterized OT-I TCR transgenic mouse and altered peptide ligand (APL) system (OVA$_{257–264}$; N4: SIINFEKL & G4: SIIGFEKL). G4 peptide is an OVA variant peptide with a single amino acid change at the highly exposed

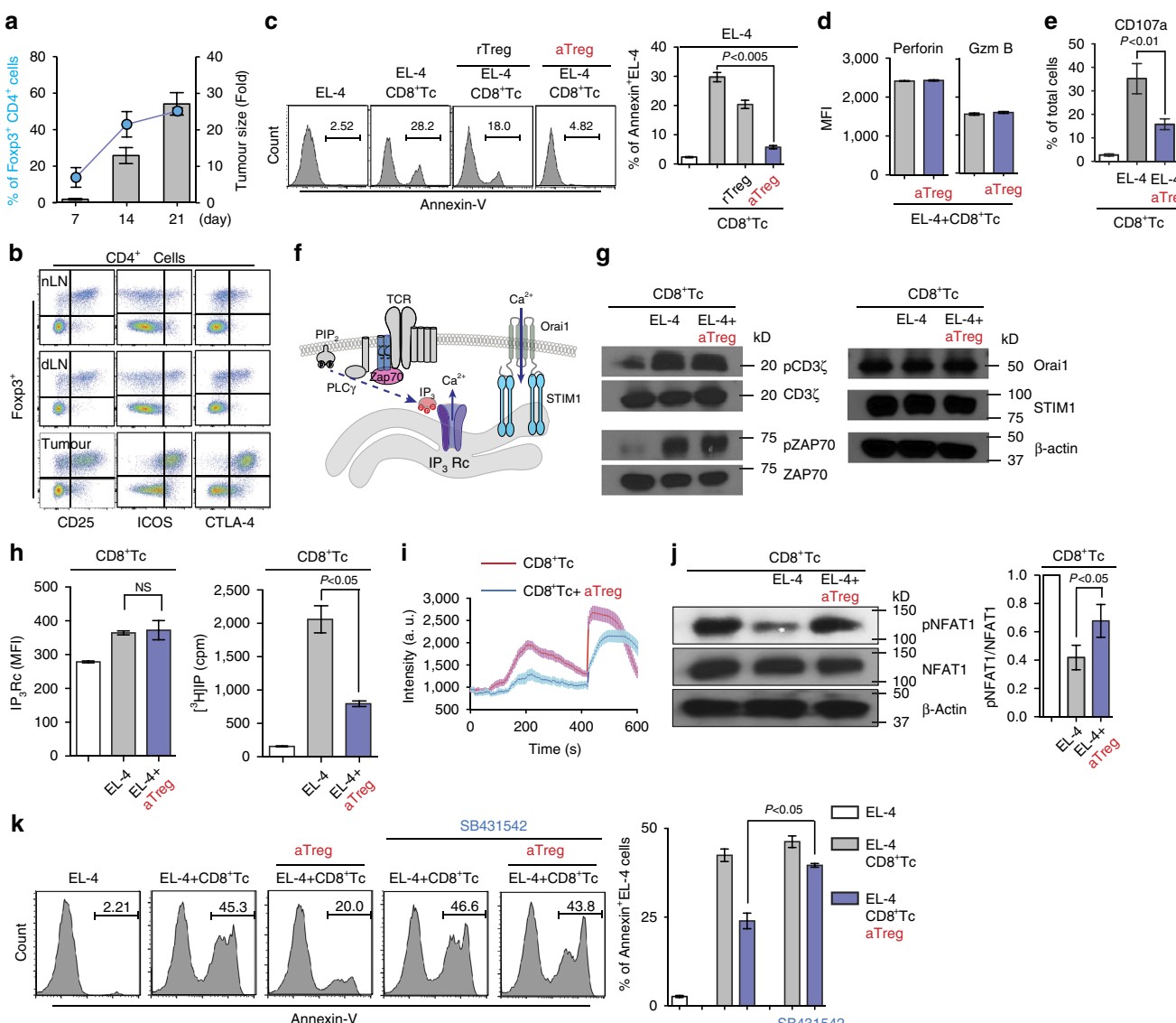

**Figure 1 | Tregs suppress CTL-mediated cytotoxicity by downregulating intracellular calcium signals.** (**a**) B16F10 melanoma cells were intradermally injected into the ear skin. Tumour growth (bar) and Treg frequency (dot) were monitored. $n > 5$ mice per group. (**b**) Phenotypic characteristics of tumour infiltrating CD4$^+$ Foxp3$^+$ Treg cells. (**c**) CTL-mediated tumour killing assay. OT-I CTLs were incubated for 5 h with OVA peptide-loaded EL-4 cells at a 1:1 ratio in the presence of rTregs or aTregs. Apoptotic cells were stained for Annexin V ($n = 3$). (**d,e**) Expression of perforin, granzyme B and CD107a on OT-I CTLs incubated with OVA-pulsed EL-4 at a 1:1 ratio in the absence or presence of aTregs was measure by flow cytometry ($n = 3$). (**f**) Schematic for intracellular Ca$^{2+}$ signals in CTL. Stimulation of TCR triggers Inositol 1,4,5 trisphosphate (IP$_3$) production via Zap70 and PLCγ activation. IP$_3$ binds to the IP$_3$ receptor (IP$_3$ Rc) on the ER membrane to empty the ER Ca$^{2+}$ store. ER Ca$^{2+}$ depletion induces STIM1 oligomerization leading to CRAC channel (Orai1) opening and (Ca$^{2+}$)$_i$ increase. (**g**) Western blot analysis of TCR signaling proteins in OT-I CTLs. OT-I CTLs were sorted after incubation with OVA-loaded EL-4 cells in the absence or presence of aTregs for 5 h. (**h**) Flow cytometry analysis of intracellular IP$_3$ receptor expression (left) and total inositol phosphate (IP) production (right) measured in OT-I CTLs ($n = 3$). (**i**) Maximum intracellular calcium release was measured by Fluo-4 intensity in OT-I CTLs after stimulation with anti-CD3 and anti-CD28 Abs followed by additional ionomycin (1 μM) stimulation at 400 s (OT-I CTL alone, $n = 19$; OT-I CTL + aTreg, $n = 13$). (**j**) Cells were further analysed by western blot (4-12% SDS–PAGE) for NFAT1 phosphorylation ($n = 3$). (**k**) OT-I CTLs were cocultured with aTreg in the absence or presence of SB431542 for 16 h. PKH-26 labeled OVA peptide-loaded EL-4 cells were added to the cells for 5 h and apoptotic EL-4 cells were stained for Annexin V ($n = 3$). Throughout, data are the mean ± s.e.m. NS: non-significant; $P < 0.01$; $P < 0.05$; $P < 0.005$; by two-tailed Student's $t$-test.

TCR contact sites on the pMHC complex and thus shows weaker affinities to TCR without altering the peptide affinity for MHC class I (Fig. 2a)[29]. Ionomycin treatment of OT-I CD8$^+$ Tc significantly increased CD8$^+$ T cell activation, cytokine production and degranulation in response to the weak-affinity antigen G4 (Fig. 2b–d, Supplementary Fig. 2). Consistently, ionomycin treatment improved the killing of G4-loaded EL-4 target cells to a level close to that achieved against a high-affinity antigen (N4)-loaded EL-4 cell (Fig. 2e).

Local immunosuppressive cells, such as Tregs, can impair the effector functions of CTLs by inhibiting lytic granule release after recognition of the target cell[12,30]. Therefore, our data led us to hypothesize that boosting intracellular Ca$^{2+}$ signals in CTLs may augment the lytic granule-dependent killing of target cells, even under the strong suppression by Tregs. To test this hypothesis, we co-cultured N4 peptide-loaded EL-4 cells with OT-I CD8$^+$ Tc in the presence of activated effector Tregs (aTregs). Preincubation of CD8$^+$ Tc with aTreg significantly reduced the tumouricidal

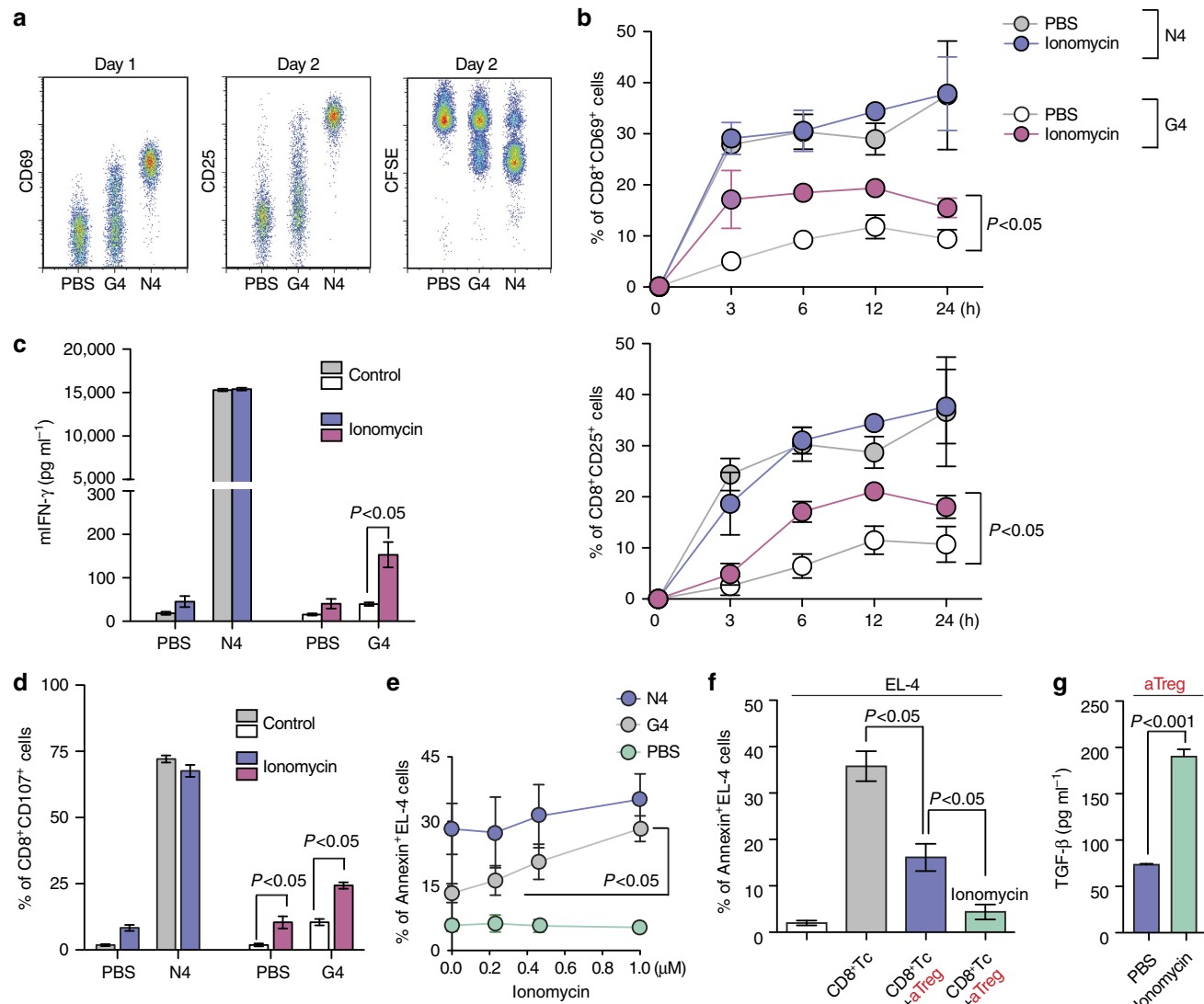

**Figure 2 | The effects of increased intracellular $Ca^{2+}$ on CTL effector functions.** (**a,b**) Proliferation (CFSE) and expression of CD69 and CD25 on OT-I $CD8^+$ T cells after activation with SIINFEKL (N4) peptide, SIIGFEKL (G4) peptide or PBS, in the absence or presence of ionomycin ($1 \mu M$, $n = 3$). (**c,d**) IFN-$\gamma$ secretion (measured by ELISA) and CD107a expression (measured by flow cytometry) on OT-I $CD8^+$ T cells activated with N4 peptide, G4 peptide or PBS, in the absence or presence of ionomycin ($1 \mu M$, $n = 3$). (**e**) N4 peptide, G4 peptide, or PBS loaded target tumour cell killing in the presence of various concentrations of ionomycin ($1 \mu M$, $n = 3$). (**f**) Inhibition of target tumour cell (N4-loaded) killing by aTreg in the absence or presence of ionomycin ($1 \mu M$, $n = 3$). (**g**) TGF-$\beta 1$ secretion by aTreg with or without ionomycin treatment for 6 h, measured by ELISA of culture supernatants ($n = 3$). Throughout, data are the mean ± s.e.m. (**b,e**) Data were analysed with One-Way ANOVA with a Bonferonni post-test. (**c,d,f,g**) Data were analysed with Student's $t$-test.

effects (Fig. 2f). However, unlike our prediction, we failed to detect any improved cytotoxic activity of $CD8^+$ Tc after stimulation with ionomycin. Surprisingly, addition of ionomycin in the assay further suppressed $CD8^+$ Tc-mediated target cell killing (Fig. 2f). This suppression in the $CD8^+$ Tc response by ionomycin treatment was likely due to the simultaneous activation of Treg functions, as measured by the increased total TGF-$\beta$ release from Tregs in the presence of ionomycin (Fig. 2g). Our data strongly suggest that delivery of non-specific $Ca^{2+}$ agonists to the tumour site will not provide the expected level of antitumour cytotoxicity by $CD8^+$ Tc. Instead, it may cause aberrant activation of local immune suppressive cells and thus a stronger suppression of $CD8^+$ Tc functions. Therefore, a more targeted and deliberate approach to selectively boost $Ca^{2+}$ signals only in $CD8^+$ Tc at the tumour site is needed.

**Optical control of intracellular $Ca^{2+}$ signal in $CD8^+$ T cells.** In hippocampal neurons, expression of CatCh, a new variant of channelrhodopsin, showed an accelerated membrane $Ca^{2+}$ permeability, with 70-fold greater light sensitivity compared to that of wild-type channelrhodopsin 2, resulting in superior optogenetic control of intracellular $Ca^{2+}$ influx[31]. In this study, we used CatCh to selectively deliver $Ca^{2+}$ activation signals only to the adoptively transferred CTLs *in vivo*, without interfering with endogenous $Ca^{2+}$ signals in other cell types in the tumour microenvironment. To test the specific $Ca^{2+}$ signals controlled by CatCh, we imaged $(Ca^{2+})_i$ in HEK293 cells transfected with CatCh. Fluorescence imaging of $(Ca^{2+})_i$ demonstrated that stimulation with green light ($488 \pm 10$ nm, $4.00$ mW) was sufficient to drive prominent $(Ca^{2+})_i$ signals in CatCh-expressing cells but not in WT control cells, indicating the

functional expression of CatCh (Fig. 3a; Supplementary Movie 1). Furthermore, light stimulation of CatCh-expressing OT-I CD8$^+$ Tc was sufficient to drive prominent intracellular dephosphorylation of NFAT1 and cytokine production (IFNγ) in CatCh-expressing cells, but not in WT control cells or under dark conditions, indicating the feasibility of the remote activation of T cell Ca$^{2+}$ signaling by light stimulation (Fig. 3b,c). The ability of CatCh to control the cytotoxic functions of CD8$^+$ Tc was further confirmed by light stimulation of CatCh-expressing OT-I CD8$^+$ Tc during co-incubation with G4-loaded EL-4 target cells. Optical stimulation of CatCh-expressing OT-I CD8$^+$ Tc yielded a significant increase in target cell killing comparing to dark condition (Fig. 3d). An important advantage of CatCh is its ability to deliver highly selective Ca$^{2+}$ stimulation in CTLs and thus boost their effector functions without activating other immunosuppressive cells, such as Tregs, at the tumour site. We demonstrated this ability by light stimulation of CatCh-expressing OT-I CD8$^+$ Tc during co-incubation with N4-loaded EL-4 target cells in the presence of aTregs. Light activation of CatCh-expressing OT-I CD8$^+$ Tc significantly increased killing of target cells, allowing them to successfully overcome the Treg-mediated suppression (Fig. 3e).

TCR activation induces IP$_3$–mediated release of Ca$^{2+}$ from ER stores. After depletion of these Ca$^{2+}$ stores, the Ca$^{2+}$ sensor STIM1 activates highly selective Orai1 Ca$^{2+}$ channel. This channel is located at the plasma membrane, and is responsible for store-operated Ca$^{2+}$ entry from outside of the T cell. In contrast, CatCh is a cell-membrane calcium channel that can bypass the IP$_3$ generation and the depletion of Ca$^{2+}$ stores and induce Ca$^{2+}$ influx directly through the membrane. Therefore, the different modes of Ca$^{2+}$ signaling may induce different downstream signaling responses in T cells. To test whether induction of Ca$^{2+}$ influx by light stimulation of CatCh can recapitulate the physiological functions of Orai1 channels during CTL killing, we generated T cell specific Orai1 conditional knockout (KO) mice by crossing Orai1$^{fl/fl}$ mice[32–35] to Cd4-Cre mice. Deletion of Orai1 gene expression in CD8$^+$ T cells was validated by PCR (Supplementary Fig. 3). CD8$^+$ T cell differentiation, activation, and expression of effector molecules (perforin and granzyme B) in Orai1 KO CD8$^+$ Tc were comparable to those in WT CD8$^+$ T cells (Fig. 3f and Supplementary Fig. 4). However, lytic granule exocytosis and killing of target EL-4 cells were severely altered in Orai1 KO CD8$^+$ Tc (Fig. 3f,g). Importantly, light stimulation of CatCh-expressing Orai1 KO CD8$^+$ Tc partially restored

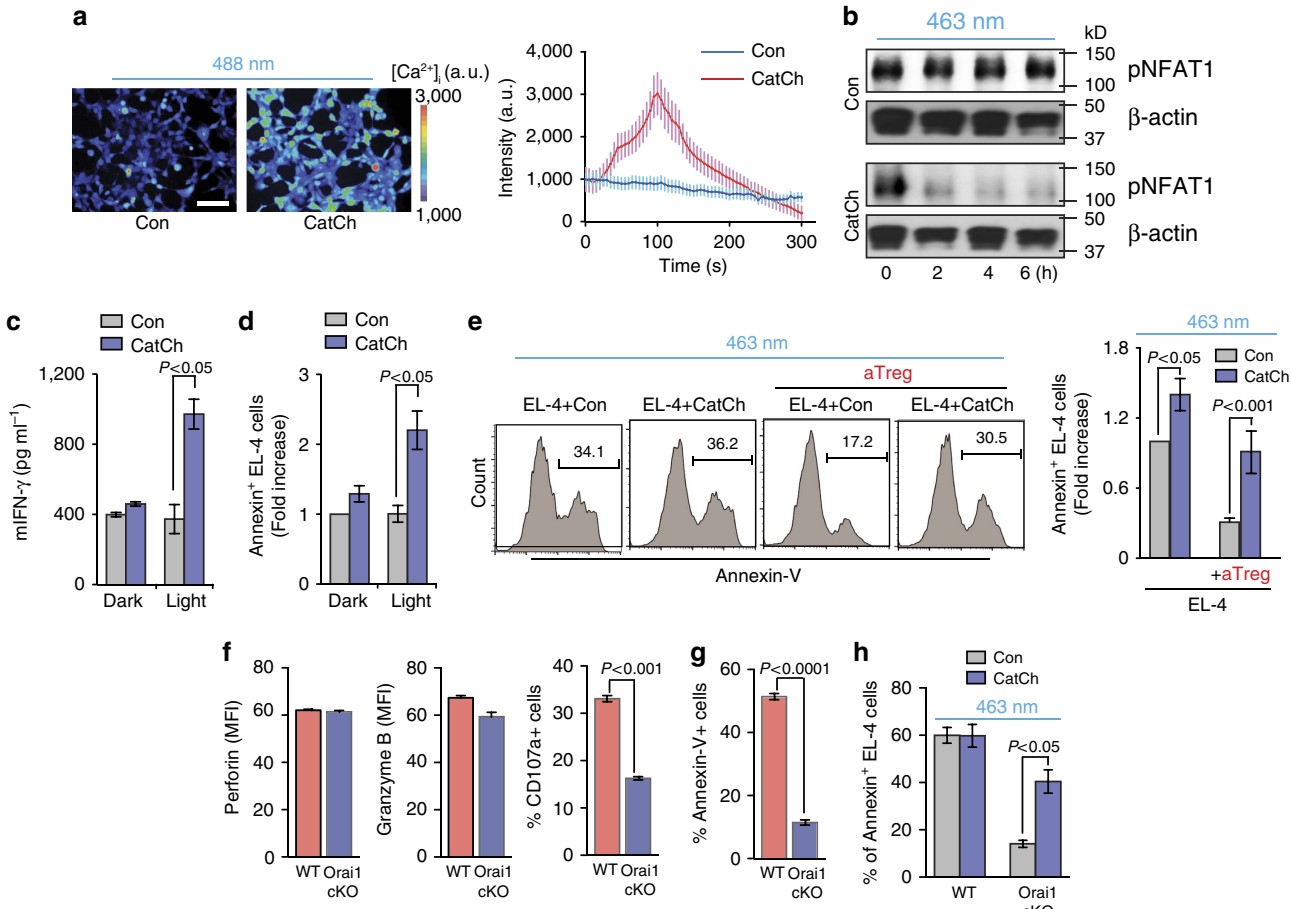

**Figure 3 | Optical control of intracellular Ca$^{2+}$ signal in CD8$^+$ T cells.** (**a**) Representative fluorescence images of Fluo-4/AM-loaded HEK293 cells during light activation. Scale bar, 100 μm. Intensity traces of mock-transfected cells (Con; $n = 16$) and cells transiently transfected with CatCh ($n = 20$). (**b**) Western blot analysis of NFAT1 phosphorylation in CatCh-expressing OT-I CD8$^+$ T cells after light stimulation for the indicated time. (**c,d**) IFN-γ secretion and cytotoxicity of CatCh-expressing OT-I CD8$^+$ T cells in the presence of G4 peptide-loaded EL-4 cells with or without light stimulation ($n = 3$). (**e**) Killing of N4 peptide-loaded EL-4 cells after light stimulation of CatCh- or GFP-expressing OT-I CD8$^+$ T cells in the presence or absence of aTregs ($n = 3$). (**f**) Expression of perforin, granzyme B and CD107a on WT or Orai1-cKO CD8$^+$ T cells after incubation with N4 peptide-loaded EL-4 cells at a 1:1 ratio for 6 h ($n = 3$). (**g**) Killing of N4 peptide loaded EL-4 cells by WT or Orai1-cKO CD8$^+$ T cells ($n = 6$). (**h**) Killing of N4 peptide loaded EL-4 cells by CatCh-expressing WT or Orai1-cKO CD8$^+$ T cells with light stimulation (463 nm) ($n = 3$). Throughout, data are the mean ± s.e.m. and were analysed by two-tailed Student's t-test.

the cytotoxic function of CTLs (Fig. 3h). These results, in combination with the earlier Treg data, support the conclusion that CatCh can be functionally expressed in CD8[+] T cells to allow photoactivatable control of $Ca^{2+}$ signals and boost T cell mediated tumour killing using remote light stimulation within the immunosuppressive tumour microenvironment.

**Optical control of CD8[+] T cell functions at the tumour sites**. To demonstrate the clinical implications of CatCh-mediated immune boosts, we examined the ability of CatCh to enhance the cytotoxicity of adoptively transferred tumour-specific CD8[+] Tc and to improve antigen-specific tumour regression. In this study, we used the Pmel-1 TCR transgenic mouse model, a well characterized mouse tumour model for low-immunogenicity, which expresses the Vα1Vβ13 TCR that recognizes an H-2D[b]-restricted epitope corresponding to amino acids 25–33 of mouse gp100 (mgp100) on the B16 melanoma cells[36]. B16 melanoma cells grow at a normal rate in Pmel-1 mice despite the presence of overwhelming numbers of mgp100-specific CD8[+] T cells[36] (Fig. 4a). Furthermore, antigen-specific vaccination with a self-antigen, mgp100 or a mgp100 altered peptide ligand (for example, human gp100; hgp100) is not sufficient to improve the antitumour effects of adoptively transferred Pmel-1 T cells against B16 tumours due to an increase in the local CD4[+]FoxP3[+] Treg cell population[36,37]. To determine whether the activation of adoptively transferred CD8[+] Tc could be enhanced in order to treat established solid tumours, we first demonstrated that light stimulation of CatCh could deliver highly selective $Ca^{2+}$ stimulation in Pmel-1 T cells and thus boost their effector functions under the Treg-mediated suppression. Light activation of CatCh-expressing Pmel-1 CD8[+] Tc significantly increased killing of hgp100$_{25-33}$ peptide (KVPRNQDWL)-loaded B16 target cells, allowing them to successfully overcome the Treg-mediated suppression (Supplementary Fig. 5A). For *in vivo* experiments, Pmel-1 CD8[+] Tc expressing CatCh were adoptively transferred into C57BL/6 mice bearing subcutaneous B16 tumours established for 7 days, followed by vaccination with mgp100$_{25-33}$ (EGSRNQDWL) or hgp100$_{25-33}$ peptide (Fig. 4b and Supplementary Fig. 5B). Subsequently, the visible and palpable tumour area was illuminated for 7 days, and tumour growth was measured for an additional 7 days without illumination. For long-term *in vivo* light exposure in freely moving animals, a battery-powered wireless blue light emitting diode (LED; 470 nm) was glued to the mouse ear skin (Fig. 4c and Supplementary Fig. 6). The peak light output during light stimulation was determined empirically at 0.1–5 mW $mm^{-2}$ (470 nm, 3.67 mW $mm^{-2}$ on average) at the surface of LED. Dark mice were treated equally for a total of 21 days without light stimulation. Localized light stimulation dramatically decreased tumour growth in mice vaccinated with hgp100$_{25-33}$ peptide (Fig. 4d). Despite the low affinity for H-2 D[b] and brief half-life of MHC complexes[38,39], light stimulation of CatCh-expressing Pmel-1 CD8[+] Tc was also marginally effective in controlling the tumour growth in mice vaccinated with the low avidity self-peptide mgp100$_{25-33}$ (Supplementary Fig. 5B). Light stimulation of mice that received GFP-transfected Pmel-1 CD8[+] Tc did not alter tumour growth, indicating that the effect of 470-nm LED light alone is not detrimental to the tumour cells (Fig. 4e). Flow cytometry analyses of B16 tumours confirmed that local light stimulation did not significantly change total T cell numbers nor intratumoural Pmel-1 CD8[+] Tc infiltration, compared to the dark mice (Fig. 4f,g). However, local light activation substantially increased the level of IFN-γ expression (Fig. 4g).

To further test whether localized light activation of CatCh-expressing CD8[+] Tc induces systemic effects and thus controls non-illuminated tumour growth at a distal secondary site, we adoptively transferred Pmel-1 CD8[+] Tc expressing CatCh into C57BL/6 mice bearing two subcutaneous B16 tumours at the ear and flank, followed by vaccination with hgp100$_{25-33}$ peptide. Subsequently, the visible and palpable tumour area at the ear was illuminated for 7 days, and tumour growth was measured at the flank. Localized light activation dramatically increased the cell surface expression of CD107a on Pmel-1 CD8[+] Tc at the illumination site (Fig. 5a), suggesting that the enhanced $Ca^{2+}$ signals in CatCh-expressing CTLs by light stimulation can not only improve the cytokine production (Fig. 4g), but also promote cytotoxic functions of CD8[+] Tc responses by inducing granule exocytosis. The enhanced local CD8[+] Tc effector functions by light stimulation at the ear significantly decreased tumour growth both at the illuminated ear and non-illuminated flank (Fig. 5b). These results strongly suggest that light stimulation of local CD8[+] Tc function may trigger systemic effects and induce anti-tumour responses outside the illumination field. Clinically, this may be an important concept, and more investigation is needed to determine the mechanism because, in view of the envisaged application of our approach, treatment of solid tumours established in their metastases is equally important.

Growing evidence suggests that Tregs do not use only one universal mechanism of immune suppression at the tumour microenvironment, but rather execute suppressive functions through several different modes[19]. Therefore it is possible that the improved antitumour activity of CatCh-expressing CD8[+] Tc by light stimulation *in vivo* may mediate a combination of multiple processes other than direct induction of lytic granule exocytosis from CTLs seen in our *in vitro* assays. Indeed, several mechanisms have been proposed for the Treg-mediated direct suppression of CD8[+] T cell anti-tumour effector functions, which include Fas/FasL-dependent T cell apoptosis[40] and suppression of effector T cells by releasing adenosine (Ado) and PGE$_2$ (refs 19,41). We addressed this possibility by light stimulation of CatCh-expressing OT-I CD8[+] Tc in the presence of soluble Fas ligand (sFasL), CGS21680 (A$_{2A}$ receptor agonist) or PGE$_2$. Light activation allowed CatCh-expressing OT-I CD8[+] Tc to successfully overcome A$_{2A}$ receptor- and PGE$_2$-mediated suppression of T cell activation in the presence of Ag-loaded APC, but failed to reverse sFasL-mediated T cell apoptosis (Fig. 5c–e).

In addition to Treg, tumour-resident myeloid-derived suppressor cells (MDSCs) can further counteract proper CTL effector functions at the tumour microenvironment. To test whether optical stimulation of $Ca^{2+}$ signals can reverse MDSC-induced CTL killing inefficiency, we stimulated CatCh-expressing OT-I CD8[+] Tc with light during co-incubation with N4-loaded EL-4 target cells in the presence of MDSCs. Light activation of CatCh-expressing OT-I CD8[+] Tc significantly increased killing of target cells, allowing them to partially overcome the MDSC-mediated suppression (Fig. 5f). Therefore, our data suggest that light stimulation of CatCh *in vivo* not only improves the lytic granule exocytosis (Fig. 5a), but also boosts suppressed CTL responses by multiple inhibitory factors derived from both Treg and MDSC.

# Discussion

In conclusion, we discovered that Treg-mediated suppression of CTL killing *in vitro* is not associated with changes in several important TCR-proximal signals but is in part mediated by TGFβ-induced inhibition of IP$_3$ production, which directly decreases intracellular $Ca^{2+}$ responses and T cell degranulation *in vitro*. Highly selective optical control of $Ca^{2+}$ signals in CTLs at the tumour site was sufficient to overcome Treg-induced immunosuppression and dramatically improve the efficacy of adoptive T cell transfer immunotherapy *in vivo*. Although direct

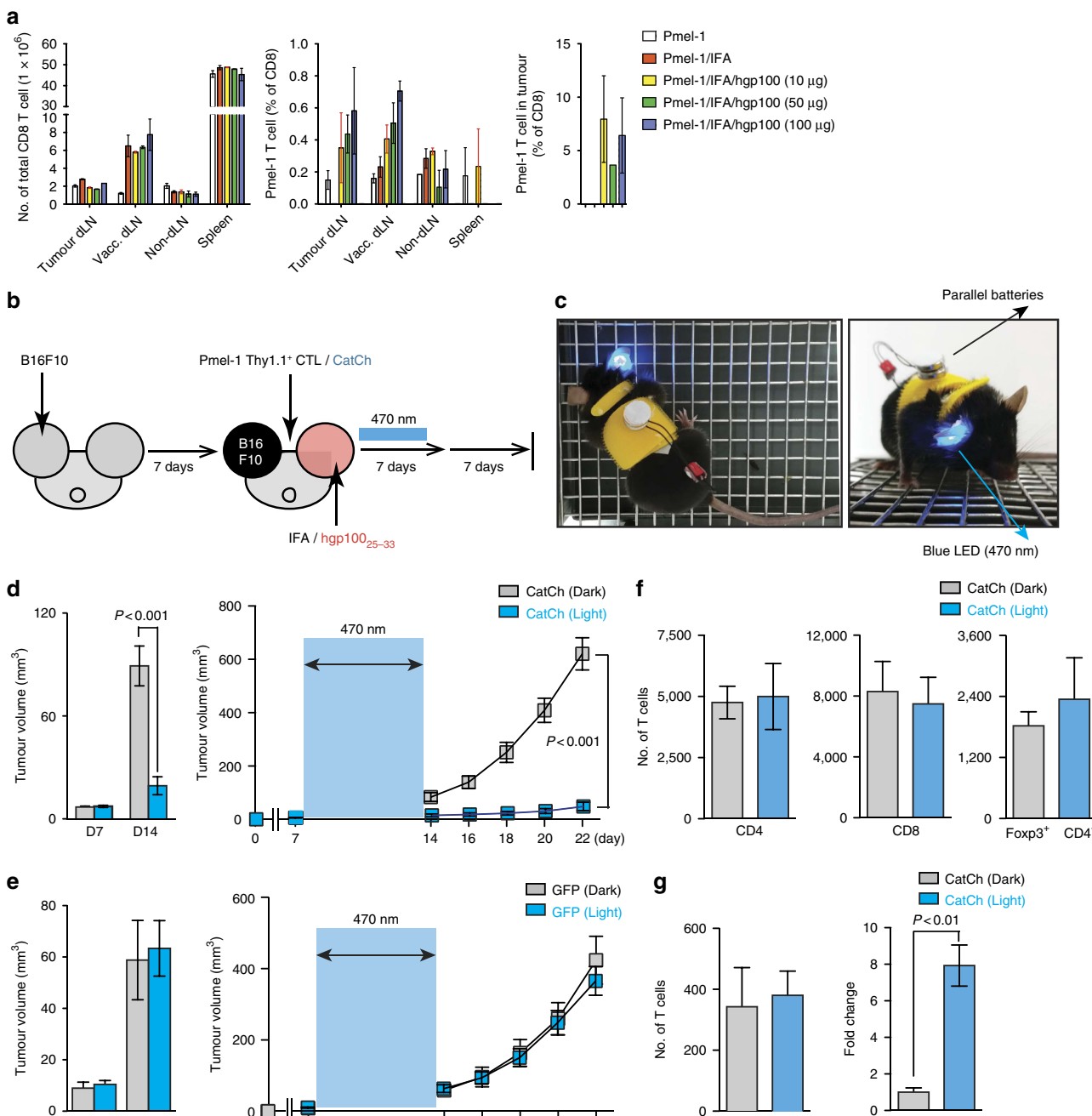

**Figure 4 | Optical control of Ca$^{2+}$ signal in adoptively transferred Pmel-1 T cells.** (**a**) Flow cytometry analysis of total T cell populations (day 14) in mice with B16 murine melanoma. Where indicated, mice were vaccinated by intradermal injection of hgp100/IFA. (**b**) Experimental design of the studies of optical control of Ca$^{2+}$ signal *in vivo*. (**c**) Freely moving mouse with battery-powered wireless LED attached on the ear skin. (**d**) Size and growth curves of B16 tumours in mice treated with adoptive transfer of CatCh-expressing Pmel-1 CD8$^+$ T cells with (optical LED + light) or without (optical LED + dark) light stimulation (light, $n = 6$; dark, $n = 6$). The light stimulation (470-nm) times are denoted with a blue box. (**e**) Size and growth curves of B16 tumours in mice treated with adoptive transfer of GFP-expressing Pmel-1 CD8$^+$ T cells with (optical LED + light) or without (optical LED + dark) light stimulation (light, $n = 7$; dark, $n = 6$). The light stimulation (470-nm) times are denoted with a blue box. (**f**) Flow cytometry analysis of total CD4$^+$ T cells, CD8$^+$ T cells, and CD4$^+$ Foxp3$^+$ Treg numbers (per $1 \times 10^6$ total cells) in B16 tumours on day 14 (after 7 days light stimulation) (light, $n = 5$; dark, $n = 8$). (**g**) Flow cytometry analysis of total Pmel-1 T cell number (per $1 \times 10^6$ total cells) (light, $n = 5$; dark, $n = 8$) and expression of IFN-γ B16 tumours was analysed by qPCR ($n = 3$). Throughout, data are the mean ± s.e.m. and were analysed by two-tailed Student's *t*-test or One-Way ANOVA with a Bonferonni post-test.

infusion of tumour-targeting CTLs or T cells that express chimeric antigen receptors has emerged as a powerful treatment option mainly for lineage restricted, non-essential targets in haematologic cancers[1], the results of clinical trials focusing on solid tumours have been much less encouraging[42] in part due to strong immunosuppressive microenvironment at tumour sites.

Enhancing anti-tumour responses using immune boosting reagents has great potential to improve cellular immunotherapy. However, as a result of their pleiotropic effects, systemic delivery often has severe side effects that have greatly limited their use. The use of light to control immune reactions avoids the need for direct physical contact with the tissue and therefore any

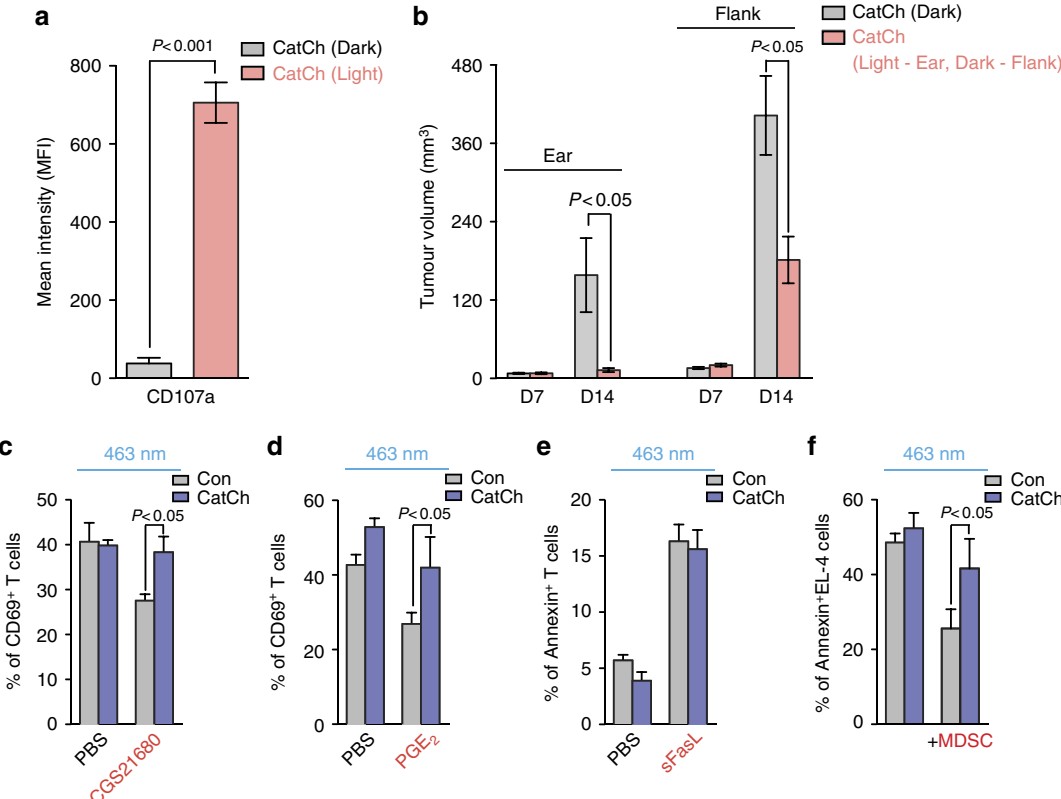

**Figure 5 | Optical control of intracellular Ca$^{2+}$ signal in CD8$^+$ T cells overcomes immunosuppression.** (**a**) $2 \times 10^5$ B16F10 cells in 10 µl PBS were intradermally injected into the ear and flank. Expression of CD107a on Pmel-1 CD8$^+$ T cells in B16 tumours in the ear with (optical LED + light) or without (optical LED + dark) light stimulation of tumours in the ear (light, $n = 5$; dark, $n = 5$). (**b**) Size of B16 tumours both in the ear and flank treated with adoptive transfer of CatCh-expressing Pmel-1 CD8$^+$ T cells with (optical LED + light) or without (optical LED + dark) light stimulation of the ear (light, $n = 5$; dark, $n = 5$). (**c**) Expression of CD69 on OT-I CD8$^+$ T cells after re-stimulation with N4 peptide (1 µg ml$^{-1}$) loaded irradiated splenocytes in the absence or presence of CGS21680 (10 µM, $n = 4$). (**d**) Expression of CD69 on OT-I CD8$^+$ T cells after re-stimulation with N4 peptide (1 µg ml$^{-1}$) loaded splenocytes in the absence or presence of PGE$_2$ (5 µM, $n = 4$). (**e**) OT-I CD8$^+$ T cells were incubated with PBS or sFasL (100 ng ml$^{-1}$) for 16 h. Apoptotic OT-I CD8$^+$ T cells were stained for Annexin V ($n = 4$). (**f**) Killing of N4 peptide-loaded EL-4 cells after light stimulation of CatCh- or GFP-expressing OT-I CD8$^+$ T cells in the absence or presence of MDSC (1:1) ($n = 6$).

interference with normal functions[43]. Importantly, light offers outstanding spatial resolution, allowing access to specific cellular subtypes and even the smallest subcellular domains. Therefore, our optogenetic approach described in this study will allow the remote control of CTL effector functions at tumour sites with outstanding specificity and temporospatial resolution.

A potential caveat of our approach is that the computed average of vertical light penetration into the mouse skin can reach only 300 µm in depth according to the light-power density profiles of our *in vivo* tissue illumination system[43]. This result suggests that our optical stimulation (Fig. 4c) can deliver a functionally active light gradient only to a tumour established in the surface of organs (for example, the skin dermis). Another significant constraint in the current system is that the LED has to be wired to the main power source to generate stable light emission. In view of the clinical application of our approach, treatment of solid tumours established in deep tissues and/or their metastases may be equally important. Recently several groups developed fully wireless and implantable optogenetic stimulation tools[44,45]. These small size optoelectronics consists of a power receiving coil, circuit and LED and can be fully implanted into deep tissues. Importantly, this miniaturized wireless device allows subjects to move freely. It could be the subject of a future study to determine whether this completely wireless, ultrathin, biocompatible LED can deliver light to a tumour established in patient's deep soft tissues and activate T cells at the tumour site.

## Methods

**Mice.** C57BL/6J, TCR-transgenic OT-I mice and CD4-cre mice were purchased from the Jackson Laboratory. The Orai1 floxed (C57BL/6 background) mice were a generous gift from Drs Yousang Gwack and Sonal Srikanth. Orai1$^{fl/fl}$ mice were bred to CD4-Cre mice to yield T cell-specific Orai1–deficient mice. Orai1$^{fl/fl}$CD4-Cre mice were bred to TCR-transgenic OT-I mice to yield Orai1$^{fl/fl}$CD4-Cre OT-I mice. The mice were housed under pathogen-free conditions. All mouse experiments were approved by the University Committee on Animal Resources at the University of Rochester.

**Cell lines culture.** EL-4 mouse lymphoma cell line (American Type Culture Collection) and B16F10 mouse melanoma cell line (American Type Culture Collection) were cultured in DMEM supplemented with 10% FCS and penicillin-streptomycin. Around 293 T cells (American Type Culture Collection) and Phoenix cells (American Type Culture Collection) were cultured in DMEM supplemented with 10% FBS, 100 U ml$^{-1}$ penicillin, 100 µg ml$^{-1}$ streptomycin, 2 mM L-glutamine, 20 mM HEPES buffer, 1% MEM Non-Essential Amino Acid and 50 µM β-mercaptoethanol. Cells lines were tested for mycoplasma contamination by PCR analysis.

**B16F10 tumour model.** Approximately $2 \times 10^5$ B16F10 cells in 10 µl PBS were intradermally injected into one ear pinna of a recipient C57BL/6 mouse. Twelve weeks old mice of both genders were used in this study. Tumour growth was monitored every week from day 7 after tumour injection. Tumour volume was calculated as width$^2$ × length × 0.52. The mice were killed on days 7, 14 and 21 post tumour injection. To analyse tumour samples, mouse ear tumours were cut into small pieces and digested with collagenase/dispase (Roche, 10269638001) at 37 °C for 30 min.

**Antibodies.** The following antibodies used for flow cytometry were purchased from eBioscience, BD Biosciences or BioLegend: V450-conjugated anti-CD4 (560468; 1/400, BD Biosciences), FITC-conjugated anti-CD8 (11-0081-81; 1/200, eBioscience), PE-conjugated anti-Thy1.1 (12-0900-83; 1/200, eBioscience), APC-conjugated anti-Foxp3 (17-5773-80A; 1/50, eBioscience), PE-conjugated anti-CD25 (12-0251-81B; 1/100, eBioscience), PE/Cy5- conjugated anti-ICOS (15-9942-81; 1/200, eBioscience), PE/Cy7- conjugated anti-CTLA-4 (106313; 1/100, BioLegend), APC-conjugated Annexin-V (550474; 1/25, BD Biosciences), APC-conjugated anti-Perforin (17-9392-80; 1/100, eBioscience), PE/Cy7-conjugated anti-Granzyme B (25-8898-82; 1/100, eBioscience), Alexa-Fluor 647-conjugated anti-CD107a (121610; 1/100, BioLegend), PE/Cy7-Conjugated anti-CD107a and PE/Cy7-conjugated anti-CD69 (104511; 1/100, BioLegend). The following antibodies were used for western blot analysis: anti-phospho-CD3-$\zeta$ (SAB4200334; 1:5,000, Sigma), anti-CD3-$\zeta$ (MA1-10188; 1/2000, Invitrogen), anti-phospho-ZAP-70 (#2717; 1/2,000, Cell Signaling), anti-ZAP-70 (#2705; 1/2,000, Cell Signaling), anti-Orai1 (PM-5207; 1/1,000, ProSci), anti-Stim1 (#5668; 1/2,000, Cell Signaling), anti-$\beta$-Actin (A2228; 1/5,000, Sigma), anti-phospho-NFAT1 (SC-32994; 1/2,000, Santa Cruz) and anti-NFAT1 (#5862; 1/2,000, Cell Signaling). Anti-CD3 (553057; 10 $\mu$g ml$^{-1}$, BD Biosciences) and anti-CD28 (#102102; 10 $\mu$g ml$^{-1}$, BioLegend) were used for calcium imaging and the antibodies were cross-linked with goat anti-hamster IgG ( #855397; 5 $\mu$g ml$^{-1}$, MP Biomedicals).

**T cell purification and activation.** OT-I CD8$^+$ T cells were purified from single-cell suspensions of lymph nodes of OT-I mice. Single-cell suspensions were prepared by mechanical disruption using a cell strainer. OT-I CD8$^+$ T cells were then enriched by magnetic-bead depletion with anti-mouse MHC Class II antibody (M5/114) and anti-mouse CD4 antibody (GK1.5), followed by Low-Tox-M Rabbit Complement (Cedarlane CL3051) and sheep anti-rat IgG magnetic beads (Invitrogen11035). For effector T cell differentiation, cells were stimulated with OVA peptide (4 $\mu$g ml$^{-1}$) and irradiated splenocytes for 5 days in IL-2 (50 U ml$^{-1}$) containing media. Regulatory T cells were purified from single-cell suspensions of lymph nodes and spleens of 8-12 week-old C57BL/6J mice and enriched with a CD4$^+$CD25$^+$ Regulatory T Cell Isolation Kit (Miltenyi Biotec, 130-091-041) according to the manufacturer's protocol. To generate activated Tregs, purified Tregs were stimulated with plate-bound anti-CD3 (3 $\mu$g ml$^{-1}$) and anti-CD28 (3 $\mu$g ml$^{-1}$) for 48 h in IL-2 (20 U ml$^{-1}$) containing media.

**Cytotoxicity assay.** OT-I CTLs were co-cultured without with or without Tregs in the presence of IL-2 (20 U ml$^{-1}$) for 16 h. To measure OT-I CTL cytotoxicity, fluorescent dye (CFSE or PKH-26)–labeled target EL4 cells were pulsed with Ova peptide (N4 or G4) for 2 h. After washing EL-4 cells three times with PBS, EL-4 cells were added to OT-I CTLs with or without Tregs. After 5 h, cytotoxicity was measured by flow cytometry after Annexin-V staining. MDSCs were isolated from solid tumours using Myeloid-Derived Suppressor Cell Isolation Kit (Miltenyi Biotec, 130-094-538). Approximately 2 × 10$^6$ EL-4 cells 100 $\mu$l PBS were injected subcutaneously in the flank of recipient C57BL/6 mouse. After 24 days, tumours were chopped into small pieces using scissor and digested with collagenase/dispase (3 mg ml$^{-1}$) for 1 h at 37 °C. Single cells were prepared by filtering through a 70 $\mu$m cell strainer. Gr-1$^+$ cells were isolated by using biotinylated anti–Gr-1 antibody and anti-biotin microbeads with LS columns (Miltenyi Biotec, 130-042-401).

**Western blot.** CFSE-labeled OT-I CTLs were isolated using fluorescence-activated cell sorting (FACS) from unlabeled EL-4 and aTreg in co-culture experiments. For protein extraction, cells were lysed in RIPA buffer (Thermo Scientific, #89900) and 1 × Halt protease & phosphatase inhibitor cocktail (Thermo Fisher Scientific, #78440). Electrophoresis was performed on PAGEr Gold Precast Gels (Lonza, #58522), and proteins were transferred to nitrocellulose membranes (Thermo scientific, #88018). After blocking with 5% nonfat dry milk, the blots were incubated overnight with the different primary antibodies used. All secondary antibodies were conjugated with horseradish peroxidase (HRP). SuperSignal West pico (Thermo scientific, #32106) and Supersignal West Femto (Pierce, #34095) were used to detect HRP on immunoblots with X-ray film (Pierce, #34090). Films were scanned using a LiDE 210 Scanner (Canon). Important uncropped western blots are shown in Supplementary Fig. 7.

**Inositol phosphate assay.** After coculture with or without aTregs, OT-I CTLs were labeled by adding 4$\mu$Ci of [$^3$H] inositol for 24 h in inositol-free F-10 media. After labeling, LiCl was added directly to the labeling media at a final concentration of 10 mM. After 10 min, peptide pulsed EL-4 were added to the plate. The final volume of each well was 1 ml. The plates were placed back in to the incubator at 37 °C for 30 min. The cells were then washed were washed twice with cold PBS. Ice-cold 50 mM formic acid, 1 ml, was added to the cells, which were placed on ice for 30 min. After the incubation, the lysates were applied to Dowex AX1-X8 columns and allowed to flow all the way through the column. The columns were washed with 50 and 100 mM formic acid, followed by elution of the inositol phosphate (IP)-containing fraction with 3 ml of 1.2 M ammonium formate/0.1 M formic acid. The eluted fraction was mixed with 10 ml of scintillation fluid and measured by liquid scintillation counting.

**Calcium imaging.** HEK293T cells were transfected with the pcDNA3.1-CatCh-mCherry plasmid using Lipofectamine 2000 (Invitrogen, 11668-030) on a Delta T culture dish (Bioptechs, 04200417C). PKH-26-labeled OT-I CTLs were cocultured with or without pre-activated Tregs before TCR stimulation. To measure intracellular calcium release, the cells were loaded with 2 $\mu$g ml$^{-1}$ Fluo-4 AM (Molecular Probes) at 37 °C for 30 min. The cells were resuspended in Leibovitz's L15 medium (Gibco) containing 2 mg ml$^{-1}$ glucose and mounted on Delta T culture dish (Bioptech). To excite Fluo-4 AM and stimulate CatCh, the cells were imaged under FITC fluorescence filter for 15 min. Later, 2 $\mu$M ionomycin (Invitrogen) was added to induce the release of maximum intracellular calcium.

**CatCh expression in T Cells.** CatCh sequence was cloned into the pMIGR-GFP vector. CatCh and GFP control retroviruses were generated using the Phoenix-ecotropic packaging cell line. For retroviral transductions, Phoenix cells were transfected with the above plasmids to produce retroviruses using the calcium phosphate transfection method. Virus-containing supernatant was collected at 2 and 3 days after transfection. OT-I CD8$^+$ T cells or Pmel-1 CD8$^+$ T cells were transduced on day 1 after activation in the presence of 8 mg ml$^{-1}$ polybrene. The cells were sorted based on GFP expression.

**Light stimulation.** Approximately 2 × 10$^5$ B16F10 cells in 10 $\mu$l PBS were intradermally injected into one ear pinna of a recipient C57BL/6 mouse. Tumour size was measured at day 7, and mice with a similar size tumour were selected for experiments. Mice were vaccinated by intradermal injection of mouse ear skin with 10 $\mu$l PBS/IFA (Incomplete Freund's adjuvant) emulsion containing 10 $\mu$g of hgp100$_{25-33}$ peptide (KVPRNQDWL). Then, 2 × 10$^6$ CatCh-expressing Pmel-1 CTLs or GFP-expressing Pmel-1 CTLs were injected either via tail vein or retro-orbital injection. Blue LED emitters (Future electronics, LXML-PB01-0030) were attached to the tumour site with glue. Lithium batteries were connected to blue LED emitters and replaced daily. The design of the Battery-powered wireless LED is described in Supplementary Fig. 5.

For *in vitro* light stimulation, an optical fiber was installed in CO$_2$ incubator. The fiber was coupled to an LED system (Doric Lens, LEDRV_2CH_1000) through a blue LED module (Doric Lens, LEDC1-B_FC). The peak light output during 463-nm light stimulation was estimated to be 17 mW cm$^{-2}$ at the tip of the optic fiber. Blue light pulses (1 Hz, 500 ms on, 500 ms off) were delivered by Optogenetics TTL Pulse generator (Doric Lens, OTPG_4). The cells were cultured in glass bottom 96-well plate (MatTek, P96G-1.5-5- F) for light illumination.

**RT-PCR.** Total RNA was extracted from collagenase/dispase-treated tumour samples using RNA isolation kit (Qiagen, #74104). Complementary DNA(cDNA) was synthesized from total RNA using Superscript III First-Strand cDNA synthesis kit (Invitrogen, #18080051). Real-time PCR was performed using an iCycler IQ5 system and SsoFast EvaGreen Supermix (Bio-Rad, #1725201) using the following primer pairs: GAPDH (forward: 5′-CATGGCCTTCCGTGTTCCTA-3′ and reverse: 5′-CCTGCTTCACCACCTTCTTGAT-3′), perforin (forward: 5′-GAGAA-GACCTATCAGGACCA-3′ and reverse: 5′-AGCCTGTGGTAAGCATG-3′), granzyme B (forward: 5′-CCTCCTGCTACTGCTGAC-3′ and reverse: 5′-GTCAGCACAAAGTCCTCTC-3′), IL-2 (forward: 5′-CCTGAGCAGGATGGA-GAATTACA-3′ and reverse: 5′-TCCAGAACATGCCGCAGAG-3′), IFN-g (forward: 5′-GGATGCATTCATGAGTATTGC-3′ and reverse: 5′-CCTTTTCCGCTTCCTGAGG-3′) and TNF-$\alpha$ (forward: 5′-GACGTGGAACT-GGCAGAAGAG-3′ and reverse: 5′-TTGGTGGTTTGTGAGTGTGAG-3′). $\beta$-actin (forward: 5′-GGCTGTATTCCCCTCCATCG-3′ and reverse: 5′-CCAGTTGG-TAACAATGCCATGT-3′) and Orai1 (forward: 5′-GATCGGCCA-GAGTTACTCCG-3′ and reverse: 5′-TGGGTAGTCATGGTCTGTGTC-3′).

**Statistical analysis.** All statistical tests were done with GraphPad Prism and Jmp Software (SAS). Statistical analysis was performed using One-Way ANOVA with a Bonferonni post-test, unpaired $t$-test, and Mann–Whitney when appropriate. Differences were considered significant when $P$ values were <0.05.

**Data availability.** The authors declare that all the data supporting the findings of this study are available within the article and its Supplementary Information files and from the corresponding author on reasonable request.

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

## Acknowledgements

We thank Eric Harrower, Emma Guilfoyle and Edith Lord for reagents and their technical assistance. We thank Ernst Bamberg for CatCh cDNA. This project was financially supported through grants from the National Institute of Health (CA194969 to M.K. and F31AI112257 to T.C.) and the Korean Federation of Science and Technology Societies (Brain Pool Program 171S-4-3-1808 to M.K and C.-D.J.).

## Author contributions

K.-D.K. conducted most of the experiments and performed the statistical analysis of the data; S.B., T.C., R.G.d.R., A.V.S. and H.N. performed both in vitro and in vivo experiments and helped in the data analysis; C.-D.J. helped with designs of the recombinant DNA constructs; B.P., W.J. and T.-I.K. designed the wireless battery powered LED system. M.K. conceived, designed and directed the study. K.-D.K. and M.K. wrote the manuscript with suggestions from all authors.

## Additional information

**Competing interests:** The authors declare no competing financial interests.

