## [Peer Review File · Nature Communications]

Reviewers' comments:

Reviewer #1 (Remarks to the Author):

Figure 4d,e is a well-controlled demonstration that light activation of CatCh-expressing Pmel-1 CD8+ T cells impairs tumor growth. This finding is worthy of further development. Unfortunately, the other experiments in the manuscript do not advance the finding and in some ways detract from it.

Most of the manuscript is devoted to a broad unrelated claim that "Treg-mediated suppression of CTL killing ... is mainly mediated by TGF β -induced inhibition of IP3 production". The claim is not essential to the experiment of Figure 4d. Nor is it thoroughly investigated and established, even for the in vitro system used here. Moreover Tregs utilize many mechanisms to damp down immune responses, as noted in the text, and a single in vitro assay cannot fully capture the tumor microenvironment and cannot prove that there is one single mechanism by which Tregs limit CD8+ T cell function in vivo.

In Figure 3e, light significantly increases EL-4 killing whether or not activated Tregs are present. This casts a shadow on the specific claim that light-activated calcium influx is overcoming an inhibitory Treg effect in Figure 4d, rather than simply enhancing the CD8+ T cell response.

Many factors limit tumor killing, including Tregs, MDSCs, and low MHC expression on tumor cells, and the dominant factors can be different in different tumors. The ability to enhance calcium signaling is potentially useful in cases of diminished CD8 T cell response that arise from a variety of causes— although this remains to be proven— and the authors might profitably pursue this angle.

The manuscript is marred by numerous incorrect claims: for example, that tumor neo-epitopes are inherently weak; that ionomycin at concentrations ordinarily used in T cell experiments acts through release of IP3; and citing Marangoni et al. in support of the claim that "Tregs have been shown to directly impair CD8+ T cell effector functions by compromising the release of lytic granules".

Reviewer #2 (Remarks to the Author):

is is technical study that demonstrate that increasing calcium signaling in T cells enhance their killing capacity in vivo and suggest a new and inovative way to enhance antitumor immunity by render cell resistance to treg. The concept is novel the data are well control and convincing.

I recommand to test the efficacy in vivo in another type of cancer to generalize the observation (such as LLCova or MC38ova for example)

It could be interesting to test similar therapy with vaccination with normal peptide and not altered peptide

Reviewer #3 (Remarks to the Author):

This article builds on a number earlier observations showing that Tregs rapidly reduce Ca(2+) influx and downstream signaling (Schwarz JI 2013), the role of TGF β in this (Asano JI 2008)adn a series of older articles showing that by using Bryostatin and Ionomycin (to induce the influx of

Ca²⁺) improves the activation of T cells (e.g. Mc Grath Am J Surg 1992, Bear CII 2001, Le CII 2009). However, Kim et al. show how Tregs actually impair Ca²⁺ signaling, why simply giving Ionomycin to restore Ca²⁺ signaling does not work in vivo as it also activates the Tregs and they come up with a new way to selectively enhance the Ca²⁺ influx and, hence the activity of adoptively transferred T cells via optical stimulation.

The data shown in Figures 1-3, and supplemental Figures 1-2 as well as the accompanying text correctly describes that Tregs are accumulating at the tumor site, that they suppress effector T cells in vitro, what the effect of suppression is, and that this is associated with impaired Ca²⁺ signaling and mediated via TGF β . Subsequently, they show conform the old literature that Ionomycin, used to enhance Ca²⁺ signaling can improve effector cells in vitro, but that in assays where Tregs are present in fact suppression is even stronger, caused by co-activation of the Tregs. Finally in figure 3, they found a way to circumvent this by the overexpression of CatCh. CatCh is a cell membrane bound Ca²⁺ channel which, when activated by light, bypasses the normal STIM1/Orai1 operated CRAC channel. Indeed light stimulation of CatCh-expressing T cells drove T cell activation and resistance to Treg mediated suppression.

In figure 4, the authors use the pmel TCR tg mouse model. These TCR tg T cell recognize a gp100 peptide on B16 tumor cells. They use this to show that in an in vivo situation, activated T cells are less bothered by the Tregs and display tumor control. For this they inject the tumor cells in the ear, attach a light diode to the ear, transfer CatCh+ pmel TCT tg cell and then show superior tumor control in mice receiving the light when compared to mice that were not treated with light. There are a number of questions related to this last figure that should be answered:

1. Formal proof is lacking that in this model pmel TCR tg cells are indeed regulated by Tregs and that this occurs via regulation of Ca²⁺ signaling. This could be easily shown in an in vitro experiment.
2. While it is very interesting that a light diode to the ear of the mice is capable of rescuing CatCh+ effector T cells from Tregs one should consider the fact that the mouse ear is very transparent (allowing the light to reach the T cells) but the human skin is much thicker. If the same technique can also be applied to subcutaneous human melanoma needs to be discussed (at least).
3. In view of the envisaged application (use it to optimize adoptive T cell transfer) one should know that this is performed in late stage 3/4 melanoma patients, meaning that they will present themselves with metastases that are in different location of the body, and certainly not all under the skin. While the skin melanoma's can be easily removed this is not the case for the metastases at other sites. Hence, the question is, if one would use this technique on one tumor (subcutaneous) would it then have also systemic effects. The authors should test this by providing the mice not only with a tumor in the ear but also a second tumor in the flank to see if this tumor then also is controlled.
4. In figure 4g the authors have measured IFN γ expression by PCR, so not directly the secretion of this molecule. While IFN γ is not directly related to increased cytotoxicity (please change this statement at page 10), there is another problem with this. The whole idea is that the Tregs do not alter the expression of cytolytic/effector molecules in the effector cells (Fig 1d) but that they prevent their secretion (Fig 1e). Hence, just to show that IFN γ expression goes up is not similar to its secretion. Although the in vivo experiment showing tumor outgrowth control is convincing that there is an effect, the authors did not formally prove that the secretion of effector molecules is rescued in vivo.

In general, it may be wise to introduce TCR-signaling and how this is influenced/mediated by Ca²⁺ signaling and the different molecules (STIM/Orai1) a bit better to the readers as currently the general public would be lost.

Reviewer #1 Comments:

Comment 1:

Most of the manuscript is devoted to a broad unrelated claim that “Treg-mediated suppression of CTL killing ... is mainly mediated by TGF β -induced inhibition of IP3 production”. The claim is not essential to the experiment of Figure 4d. Nor is it thoroughly investigated and established, even for the *in vitro* system used here. Moreover Tregs utilize many mechanisms to damp down immune responses, as noted in the text, and a single *in vitro* assay cannot fully capture the tumor microenvironment and cannot prove that there is one single mechanism by which Tregs limit CD8+ T cell function *in vivo*.

Response:

The reviewer raised a very relevant point regarding the multiple potential mechanisms by which Treg cells control immune responses at the tumor site. When exposed to target tumor cells, CD8⁺ cytotoxic T cell (CD8⁺ Tc) directly releases the cytotoxins including perforin, granzymes, and granulysin. Through the action of perforin, granzymes enter the cytoplasm of the target cell and their serine protease function activates caspase cascades that eventually lead to tumor cell apoptosis. Therefore, the release of granule contents accounts for the most direct and final step in CD8⁺ Tc effector function. Although multiple mechanisms underlying Treg-mediated immune suppression have been proposed [1], little is known about the role of Treg in regulation of the direct tumor killing process of CD8⁺ Tc. In this study, we used the CatCh receptor, which can induce Ca²⁺ influx upon light stimulation, and showed that highly selective intracellular Ca²⁺ signals in T cells can successfully reverse Treg cell-mediated suppression of the granule exocytosis and tumoricidal functions of CD8⁺ Tc.

Of course, our studies with CatCh-expressing CD8⁺ Tc do not support the conclusion that Ca²⁺ influx is the only mechanisms by which Tregs limit CD8+ T cell function *in vivo*. Any language alluding to this has been removed from the revised manuscript and we modified our discussion accordingly in the text. Instead, we fully agree with the reviewer’s comment that the improved antitumor activity of CatCh-expressing CD8⁺ Tc by light stimulation *in vivo* may be mediated by a combination of multiple processes other than direct induction of lytic granule exocytosis from CTLs seen in our *in vitro* assays. Indeed, several mechanisms have been proposed for the Treg-mediated direct suppression of CD8⁺ T cell anti-tumor effector functions, which include Fas/FasL-dependent T cell apoptosis [2] and suppression of effector T cells by releasing adenosine (Ado) and PGE₂ [1, 3]. We addressed this possibility by light stimulation of CatCh-expressing OT-I CD8⁺ Tc in the presence of soluble Fas ligand (sFasL), CGS21680 (A_{2A} receptor agonist), or PGE₂. Light activation allowed CatCh-expressing OT-I CD8⁺ Tc to successfully overcome A_{2A} receptor- and PGE₂-mediated suppression of T cell activation in the presence of Ag-loaded APC, but failed to reverse sFasL-mediated T cell apoptosis (new Fig. 5). Our data suggest that light stimulation of CatCh *in vivo* not only improves the lytic granule exocytosis (new Fig. 5a), but also boosts CTL responses, which have been suppressed by multiple inhibitory factors derived from Treg (Fig. 5c - e).

Comment 2:

In Figure 3e, light significantly increases EL-4 killing whether or not activated Tregs are present. This casts a shadow on the specific claim that light-activated calcium influx is overcoming an inhibitory Treg effect in Figure 4d. rather than simply enhancing the CD8+ T cell response.

Response:

As we discussed above (see the response to the comment #1), we agree with the reviewer's comment that the improved antitumor activity of CatCh-expressing CD8⁺ Tc by light stimulation *in vivo* may mediate a combination of multiple processes other than direct induction of lytic granule exocytosis from CTLs seen in our *in vitro* assays. Our new data in **Fig. 5 (c – f)** further supports this conclusion. We have added a statement to the text (**pg. 11 - 13**) to emphasize this point.

Comment 3:

Many factors limit tumor killing, including Tregs, MDSCs, and low MHC expression on tumor cells, and the dominant factors can be different in different tumors. The ability to enhance calcium signaling is potentially useful in cases of diminished CD8 T cell response that arise from a variety of causes—although this remains to be proven— and the authors might profitably pursue this angle.

Response:

We thank the reviewer for the comment and completely agree with this opinion.

In addition to preventing T cell trafficking, many tumors can downregulate antigen presentation through reduced antigen processing or MHC expression to effectively render T cells “blind” to their presence. Furthermore, proper CTL effector functions are counteracted by tumor-resident suppressive immune cells, including CD4⁺CD25⁺Foxp3⁺ regulatory T cells (Tregs) and myeloid-derived suppressor cells (MDSCs). Our data in **Fig. 5 (c – f)** suggest that an increase in the intracellular Ca²⁺ concentration can enhance the effector functions of CTLs under various suppressive environments in response to low-affinity altered peptide ligands (APLs) for TCRs and to other immune suppressive molecules. We have added a statement to the text (**pg. 11 -13**) to emphasize this idea.

Comment 4:

The manuscript is marred by numerous incorrect claims: for example, that tumor neo-epitopes are inherently weak; that ionomycin at concentrations ordinarily used in T cell experiments acts through release of IP3; and citing Marangoni et al. in support of the claim that “Tregs have been shown to directly impair CD8+ T cell effector functions by compromising the release of lytic granules”.

Response:

We have corrected the errors and modified the text accordingly.

Reviewer #2 Comments:

Comment 1:

I recommend to test the efficacy in vivo in another type of cancer to generalize the observation (such as LLCova or MC38ova for example).

Response:

The reviewer raised a very relevant point regarding potential applications of our optogenetic approach to other tumor types. Although we think this is an excellent suggestion to corroborate our conclusion and where we feel we need next go, at this time we are not fully competent to perform the same experiments with different tumor models within the revising period. Therefore, as the editor requested (**“our editorial opinion is that extending the applicability of your system to other tumor types, despite being of interest, is beyond the scope of the current manuscript and thus we would not consider it essential.”**), we decided to save this for a future pre-clinical study to demonstrate a broad application of our technology.

Comment 2:

It could be interesting to test similar therapy with vaccination with normal peptide and not

altered peptide

Response:

As suggested, we performed new experiments with a normal peptide (mgp100₂₅₋₃₃ (EGSRNQDWL)). Pmel-1 CD8⁺ Tc expressing CatCh were adoptively transferred into C57BL/6 mice bearing subcutaneous B16 tumors established for 7 days, followed by vaccination with mgp100₂₅₋₃₃ or hgp100₂₅₋₃₃ peptide. Subsequently, the visible and palpable tumor area was illuminated for 7 days, and tumor growth was measured for an additional 7 days without illumination. Localized light stimulation dramatically decreased tumor growth in mice vaccinated with hgp100₂₅₋₃₃ peptide. **Despite the low affinity for H-2 D^b and brief half-life of MHC complexes [4, 5], light stimulation of CatCh-expressing Pmel-1 CD8⁺ Tc was also marginally effective in controlling the tumor growth in mice vaccinated with the low avidity self-peptide mgp100₂₅₋₃₃.** We have added this new data in **Suppl. Fig. 5b** and modified the text accordingly (**pg. 11**).

Reviewer #3 Comments:

Comment 1:

Formal proof is lacking that in this model pmel TCR tg cells are indeed regulated by Tregs and that this occurs via regulation of Ca²⁺ signaling. This could be easily shown in an in vitro experiment.

Response:

As suggested, we demonstrated that light stimulation of CatCh could deliver highly selective Ca^{2+} stimulation in Pmel-1 T cells and thus boost their effector functions under the Treg-mediated suppression. As shown in **Suppl. Fig. 5a (new)**, light activation of CatCh-expressing Pmel-1 CD8^+ Tc significantly increased killing of hgp100₂₅₋₃₃ peptide (KVPRNQDWL)-loaded B16 target cells target, allowing them to successfully overcome the Treg-mediated suppression (**pg. 10**).

Comment 2:

While it is very interesting that a light diode to the ear of the mice is capable of rescuing CatCh+ effector T cells from Tregs one should consider the fact that the mouse ear is very transparent (allowing the light to reach the T cells) but the human skin is much thicker. If the same technique can also be applied to subcutaneous human melanoma needs to be discussed (at least).

Response:

A potential caveat of our approach is that the computed average of vertical light penetration into the mouse skin can reach only 300 μm in depth according to the light-power density profiles of our *in vivo* tissue illumination system [6]. This result suggests that our optical stimulation (Fig. 4C) can deliver a functionally active light gradient only to a tumor established in the surface of organs (e.g. the skin dermis). Another significant constraint in the current system is that the light-emitting diode LED has to be wired to the main power source to generate stable light emission. In view of the clinical application of our approach, treatment of solid tumors established in deep tissues and/or their metastases may be equally important. Recently several groups developed fully wireless and implantable optogenetic stimulation tools [7, 8]. These small size optoelectronics consists of a power receiving coil, circuit and LED and can be fully implanted into deep tissues. Importantly, this miniaturized wireless device allows subjects to move freely. It could be the subject of a future study to determine whether this completely wireless, ultrathin, biocompatible LED can deliver light to a tumor established in patient's deep soft tissues and activate T cells at the tumor site. We have included this discussion in the revised manuscript (**pg. 14-15**).

Comment 3:

In view of the envisaged application (use it to optimize adoptive T cell transfer) one should know that this is performed in late stage 3/stage 4 melanoma patients, meaning that they will present themselves with metastases that are in different location of the body, and certainly

not all under the skin. While the skin melanoma's can be easily removed this is not the case for the metastases at other sites. Hence, the question is, if one would use this technique on one tumor (subcutaneous) would it then have also systemic effects. The authors should test this by providing the mice not only with a tumor in the ear but also a second tumor in the flank to see if this tumor then also is controlled.

Response:

This is an excellent suggestion! As suggested, we performed new experiments to test whether localized light activation of CatCh-expressing CD8⁺ Tc induces systemic effects, and thus controls non-illuminated tumor growth at a distal secondary site. We adoptively transferred Pmel-1 CD8⁺ Tc expressing CatCh into C57BL/6 mice bearing two subcutaneous B16 tumors at the ear and flank, followed by vaccination with hgp100₂₅₋₃₃ peptide. Subsequently, the visible and palpable tumor area at the ear was illuminated for 7 days, and tumor growth was measured at the flank. Localized light stimulation at the ear significantly decreased tumor growth both at the illuminated ear and non-illuminated flank. **These results suggest that light stimulation of local CD8⁺ Tc function may trigger systemic-effects and induce anti-tumor responses outside the illumination field.** We agree with the Reviewer's comment that this may be a clinically important concept and believe that more investigation is needed to determine the mechanism. We have added the new data in **Fig. 5b** and modified the text (**pg. 11 -12**) to emphasize this idea.

Comment 4:

In figure 4g the authors have measured IFN γ expression by PCR, so not directly the secretion of this molecule. While IFN γ is not directly related to increased cytotoxicity (please change this statement at page 10), there is another problem with this. The whole idea is that the Tregs do not alter the expression of cytolytic/effector molecules in the effector cells (Fig 1d) but that they prevent their secretion (Fig 1e). Hence, just to show that IFN γ expression goes up is not similar to its secretion. Although the in vivo experiment showing tumor out growth control is convincing that there is an effect, the authors did not formally prove that the secretion of effector molecules is rescued in vivo.

Response:

We agree with the reviewer and have added new data (**Fig. 5a**) and modified the text accordingly (**pg. 12**). Our data showed that **local light activation substantially increased the expression level of cell surface CD107a (Fig. 5a),** suggesting that the enhanced Ca²⁺ signals in CatCh-expressing CTLs by light stimulation can improve the cytotoxic functions of CD8⁺ Tc responses by promoting granule exocytosis.

Comment 5:

In general, it may be wise to introduce TCR-signaling and how this is influenced/mediated by Ca²⁺ signaling and the different molecules (STIM/Orai1) a bit better to the readers as currently the general public would be lost.

Response:

We agree with the Reviewer and have added a schematic of TCR-signals in **Figure 1f (new)**.

1. Whiteside, T.L., *The role of regulatory T cells in cancer immunology*. Immunotargets Ther, 2015. **4**: p. 159-71.
2. Strauss, L., C. Bergmann, and T.L. Whiteside, *Human circulating CD4⁺CD25^{high}Foxp3⁺ regulatory T cells kill autologous CD8⁺ but not CD4⁺ responder cells by Fas-mediated apoptosis*. J Immunol, 2009. **182**(3): p. 1469-80.
3. Murugaiyan, G. and B. Saha, *IL-27 in tumor immunity and immunotherapy*. Trends Mol Med, 2013. **19**(2): p. 108-16.
4. Overwijk, W.W., et al., *gp100/pmel 17 is a murine tumor rejection antigen: induction of "self"-reactive, tumoricidal T cells using high-affinity, altered peptide ligand*. J Exp Med, 1998. **188**(2): p. 277-86.
5. Gold, J.S., et al., *A single heteroclitic epitope determines cancer immunity after xenogeneic DNA immunization against a tumor differentiation antigen*. J Immunol, 2003. **170**(10): p. 5188-94.
6. Xu, Y., et al., *Optogenetic control of chemokine receptor signal and T-cell migration*. Proc Natl Acad Sci U S A, 2014.
7. Montgomery, K.L., et al., *Wirelessly powered, fully internal optogenetics for brain, spinal and peripheral circuits in mice*. Nat Methods, 2015. **12**(10): p. 969-74.
8. Kim, T.I., et al., *Injectable, cellular-scale optoelectronics with applications for wireless optogenetics*. Science, 2013. **340**(6129): p. 211-6.

REVIEWERS' COMMENTS:

Reviewer #1 (Remarks to the Author):

The revised manuscript is focused more consistently on the essential findings, and incorporates new information that strengthens its relevance to tumor immunology. I support publication of the manuscript with some revisions.

(1) The narrower viewpoint of the original submission is still reflected in some statements in this version of the manuscript.

> Introduction: "Both in vivo and in vitro assays revealed that highly selective optical control of Ca²⁺ signaling in adoptively transferred CTLs was sufficient to overcome immunosuppression at the tumor site by enhancing T cell activation, IFN- γ production and antitumor cytotoxicity, leading to a significant reduction in tumor growth in mice." The measurements of T cell activation (by CD25 and CD69 induction) and IFN- γ production are independent observations, and have not been connected mechanistically to antitumor cytotoxicity. Therefore the use of "by" in this description is not supported by the data.

> Introduction: "... by boosting T cell immune responses only at the tumor site." New evidence added to this version of the manuscript shows effects on a tumor at a distant nontargeted site.

> pp. 6-7: "natural tumor antigens in general elicit relatively less robust T cell responses" Less robust than what? Many peptides chosen randomly from a wildtype human protein would also show limited or no response. And the remainder of the sentence seems to be making the point that there are multiple mechanisms that damp down T cell responses against a tumor, which is the key point that the authors need to bring out here.

> p. 7: "augment the ... killing of target cells that express weak antigens" Here again, wouldn't it make sense to emphasize that more robust Ca²⁺ signals enhance the response in the face of a variety of suppressive mechanisms, which gives this approach more generality?

(2) In Fig. 2b-2d, the different y-axis scales for N4 and G4 samples are deceptive. The two conditions are being compared, and they should be plotted on the same scale.

(3) Text descriptions need to be accurate:

> p. 4: "... while incubation with rTreg had minimal effect on the levels of cytotoxicity (Fig. 1c)." The effect shown is a lesser effect, but it is not minimal.

> p. 6, referring to Fig. 2e: "to a level similar to" The level is closer to that for N4 samples, but there is still a clear difference.

> p. 9: "... successfully restored the cytotoxic function of CTLs (Fig. 3h)." Partially restored.

> p. 13: "... allowing them to successfully overcome the MDSC-mediated suppression (Fig. 5f)." The effect is again incomplete.

> p. 13 "Treg-mediated suppression of CTL killing is not induced by changes in TcR-proximal signals" In the conditions examined. It is an overreach to draw a general conclusion that embraces all possible effects of Tregs on CTL killing.

(4) p. 6 and Fig. 1k: The statistical comparison for the statement that suppression of cytotoxicity was decreased by the inhibitor should be between the SB431542 aTreg samples and the corresponding control samples without SB431542.

(5) Ionomycin triggers increases in cytoplasmic Ca²⁺ by transporting Ca²⁺ out of ER stores, and secondarily by the resulting opening of store-operated channels in the plasma membrane. Elevated cytoplasmic Ca²⁺ can in turn activate phospholipases, but this is a byproduct of Ca²⁺ elevation, not a mechanism. The statements on p. 5 and p. 9 implying that ionomycin acts through generation of IP₃ are misleading. The Chatila paper cited (ref. 24; Figure 5) in fact shows that chelating cytoplasmic Ca²⁺ completely blocks the PLC activity.

(6) p. 5: "granule-mediated target cell killing ... requires store-operated Ca²⁺ entry" Refs. 20 and 21 demonstrate the requirement for extracellular Ca²⁺, but they predate the definition of store-operated Ca²⁺ entry in T cells. The sentence could be brought up to date by citing additionally Pores-Fernando, A.T. and Zweifach, A. (2009) Immunol. Rev. 231, 160-173.

(7) p. 6: Refs. 30 and 31 do not show that ITK activation is PLC γ mediated. Rather, PLC γ activation is downstream of, and partially controlled by, ITK.

p. 5: Define "IP" in text or figure legend.

Fig. 1i is hard to interpret. The first peak is not purely Ca²⁺ influx as stated, since it includes Ca²⁺ released from stores. It is not clear whether the rate of rise of the second peak is affected, or only the later progression of the transient, which is likely to be influenced by feedback mechanisms. And it is not clear how Tregs might act on the response to ionomycin, if that is what the authors are proposing, and how that fits into the narrative of this manuscript.

Fig. 1j: Dephospho-NFAT1 does not migrate with phosphorylated NFAT1 at ~135 kDa. Either the gel has been run in a way that does not resolve the two bands, in which case the "135 kD" label is incorrect, or there is something else wrong with the experiment.

Reviewer #2 (Remarks to the Author):

No additional comment.

Reviewer #3 (Remarks to the Author):

I am satisfied with the reply of the authors. In principle all the necessary changes and experiments have been made or performed, respectively.

Reviewer #1 Comments:

Comment 1:

Introduction: “Both in vivo and in vitro assays revealed that highly selective optical control of Ca²⁺ signaling in adoptively transferred CTLs was sufficient to overcome immunosuppression at the tumor site by enhancing T cell activation, IFN- γ production and antitumor cytotoxicity, leading to a significant reduction in tumor growth in mice.” The measurements of T cell activation (by CD25 and CD69 induction) and IFN- γ production are independent observations, and have not been connected mechanistically to antitumor cytotoxicity. **Therefore the use of “by” in this description is not supported by the data.**

Response:

We have modified the text accordingly (pg 2). New change: “*Highly selective optical control of Ca²⁺ signaling in adoptively transferred CTLs enhanced T cell activation and IFN- γ production in vitro, leading to a significant reduction in tumor growth in mice.*”

Comment 2:

Introduction: “**... by boosting T cell immune responses only at the tumor site.**” New evidence added to this version of the manuscript shows effects on a tumor at a distant nontargeted site.

Response:

We have modified the text accordingly (pg 2). New change: “*... by boosting T cell immune responses at the tumor sites.*”

Comment 3

pp. 6-7: “**natural tumor antigens in general elicit relatively less robust T cell responses**” Less robust than what? Many peptides chosen randomly from a wildtype human protein would also show limited or no response. And the remainder of the sentence seems to be making the point that there are multiple mechanisms that damp down T cell responses against a tumor, which is the key point that the authors need to bring out here.

Response:

We agree with the reviewer and have deleted this sentence (pg 7).

Comment 4

p. 7: “**augment the ... killing of target cells that express weak antigens**” Here again, wouldn't it make sense to emphasize that more robust Ca²⁺ signals enhance the response in the face of a variety of suppressive mechanisms, which gives this approach more generality?

Response:

We agree with the reviewer and have deleted this sentence (now in pg 7).

Comment 5

In Fig. 2b-2d, the different y-axis scales for N4 and G4 samples are deceptive. The two conditions are being compared, and they should be plotted on the same scale.

Response:

We agree with the reviewer and have modified the figures accordingly (Figure 2).

Comment 6

p. 4: “... while incubation with rTreg had minimal effect on the levels of cytotoxicity (Fig. 1c).” The effect shown is a lesser effect, but it is not minimal.

Response:

As suggested, we have modified the text (pg. 5).

Comment 7

p. 6, referring to Fig. 2e: “to a level similar to” The level is closer to that for N4 samples, but there is still a clear difference.

Response:

As suggested, we have modified the text (pg. 7).

Comment 8

p. 9: "... successfully restored the cytotoxic function of CTLs (Fig. 3h)." Partially restored.

Response:

As suggested, we have modified the text (pg. 10).

Comment 9

p. 13: "... allowing them to successfully overcome the MDSC-mediated suppression (Fig. 5f)." The effect is again incomplete.

Response:

As suggested, we have modified the text (pg. 13).

Comment 9

p. 13 "Treg-mediated suppression of CTL killing is not induced by changes in TcR-proximal signals" In the conditions examined. It is an overreach to draw a general conclusion that embraces all possible effects of Tregs on CTL killing.

Response:

As suggested, we have modified the text (pg. 14).

Comment 10

p. 6 and Fig. 1k: The statistical comparison for the statement that suppression of cytotoxicity was decreased by the inhibitor should be between the SB431542 aTreg samples and the corresponding control samples without SB431542.

Response:

We agree with the reviewer and have modified the figure accordingly (Figure 1K).

Comment 11

Ionomycin triggers increases in cytoplasmic Ca²⁺ by transporting Ca²⁺ out of ER stores, and secondarily by the resulting opening of store-operated channels in the plasma

membrane. Elevated cytoplasmic Ca²⁺ can in turn activate phospholipases, but this is a byproduct of Ca²⁺ elevation, not a mechanism. **The statements on p. 5 and p. 9 implying that ionomycin acts through generation of IP3 are misleading.** The Chatila paper cited (ref. 24; Figure 5) in fact shows that chelating cytoplasmic Ca²⁺ completely blocks the PLC activity.

Response:

We agree with the reviewer and have deleted the sentences both in pg. 7 and pg.9.

Comment 12

p. 5: “granule-mediated target cell killing ... requires store-operated Ca²⁺ entry” Refs. 20 and 21 demonstrate the requirement for extracellular Ca²⁺, but they predate the definition of store-operated Ca²⁺ entry in T cells. **The sentence could be brought up to date by citing additionally Pores-Fernando, A.T. and Zweifach, A. (2009) Immunol. Rev. 231, 160-173.**

Response:

As suggested, we have added the reference (pg.5).

Comment 13

p. 6: Refs. 30 and 31 do not show that ITK activation is PLC γ mediated. Rather, PLC γ activation is downstream of, and partially controlled by, ITK.

Response:

We have corrected the error (pg. 6). New change: *“Importantly, it was shown that TGF β suppresses Ca²⁺ influx in activated T cells in part through the inhibition of interleukin-2 tyrosine kinase (ITK)-mediated PLC γ activation.”*

Comment 14

p. 5: Define “IP” in text or figure legend.

Response:

We have modified the figure legend (Fig. 1h).

Comment 15

Fig. 1i is hard to interpret. **The first peak is not purely Ca²⁺ influx as stated, since it includes Ca²⁺ released from stores.** It is not clear whether the rate of rise of the second peak is affected, or only the later progression of the transient, which is likely to be influenced by feedback mechanisms. And it is not clear how Tregs might act on the response to ionomycin, if that is what the authors are proposing, and how that fits into the narrative of this manuscript.

Response:

We agree with the reviewer and have modified the sentence (Ca²⁺ flux → intracellular Ca²⁺ responses). Furthermore, we deleted the statement indicating potential ionomycin functions through IP₃ (pg. 6).

Comment 16

Fig. 1j: Dephospho-NFAT1 does not migrate with phosphorylated NFAT1 at ~135 kDa. Either the gel has been run in a way that does not resolve the two bands, in which case the “135 kD” label is incorrect, or there is something else wrong with the experiment.

Response:

Unlike the WB condition we used previously [1] (7.5% gel with extended run time plus pan-NFAT1 Ab), immunoblotting of dephosphorylated- and phosphorylated-NFAT1 in this study was performed with 4-12% gradient SDS-PAGE gels with anti-Phospho-NFAT1 (Santa Cruz). Therefore, our WB blot condition does not resolve the two different bands (dephosphorylated-NFAT1 vs. phosphorylated-NFAT1). We highlighted this procedure both in “Method” section and figure legend (Fig. 1J).

1. Kim, K.D., et al., *ORAI1 deficiency impairs activated T cell death and enhances T cell survival*. J Immunol, 2011. **187**(7): p. 3620-30.